# Decoding Health Professionals’ Attitudes and Perceptions Towards Plant-Based Nutrition: A Narrative Review

**DOI:** 10.3390/nu17132095

**Published:** 2025-06-24

**Authors:** Judith Sempa, Priscilla Brenes, Kelly Whitehair, Lonnie Hobbs, Tandalayo Kidd

**Affiliations:** 1Department of Food, Nutrition, Dietetics, and Health, Kansas State University, 919 Mid-Campus Drive North, Manhattan, KS 66506, USA; pbrenes@ksu.edu (P.B.); stirtz@ksu.edu (K.W.); 2Department of Agricultural Economics, Kansas State University, 919 Mid-Campus Drive North, Manhattan, KS 66506, USA; lhobbs@ksu.edu; 3College of Health and Human Sciences, Iowa State University, 901 Strange Road, Ames, IA 50011, USA; tkidd@iastate.edu

**Keywords:** plant-based diets, attitudes, health professionals, obesity

## Abstract

**Background/Objectives:** The ongoing obesity epidemic remains a significant public health challenge in the U.S. Nearly one-third of adults are overweight, and nearly half of the population (42.4%) are obese. These conditions, driven by poor and unsustainable diets, are major risk factors for several chronic diseases, including heart disease, which continues to be the leading cause of death in the country. This review aims to examine existing research on health care professionals’ attitudes and perceptions of plant-based nutrition and explore how this knowledge can be utilized to promote the adoption of plant-based diets (PBDs) among Americans as an alternative to the standard American diet. **Methods:** PubMed and Web of Science databases were searched in April, 2024. Out of the 151 articles identified, 27 were deemed eligible and included in the narrative review. **Results:** Nine key themes were identified as major influences on the attitudes and behaviors of health professionals regarding PBDs. These themes were mapped with the domains of the Theoretical Domains Framework (TDF) to stratify key enablers and barriers to implementation of PBDs in routine care for patients. **Conclusions:** Key barriers to incorporating plant-based nutrition into routine care include time constraints, limited educational resources, insufficient skills, lack of multidisciplinary collaboration, and inadequate professional training. Access to evidence-based research summaries, clear guidelines, ongoing professional development, and other relevant educational resources were identified as facilitators of successfully integrating PBDs into everyday practice.

## 1. Introduction

Although obesity has been around for centuries, the obesity epidemic is a new phenomenon that continues to be an issue of public health concern in the US [1]. According to the National Institute of Health (NIH) [2], nearly one in three adults in the US are overweight, and nearly half the population (42.4%) are obese. Children are also grappling with obesity, with close to twenty percent (19.3%) of children aged two to nineteen years being obese, despite intense focus on reducing childhood obesity [3]. In the last one hundred years, the rate of obesity has increased from 3.4% to 35%, a tenfold jump [4]. Being overweight or obese are risk factors for several other chronic diseases, including diabetes, high blood pressure, stroke, certain cancers, and heart disease, which is America’s number one killer [1,5]. These chronic conditions have significant health and economic costs. According to the Centers for Disease Control and Prevention (CDC), 90% of the country’s 4.1 trillion dollar annual health expenditure is spent on chronic and mental health conditions. Obesity alone costs America’s health care system USD 173 billion annually [6].

The etiology of obesity is complex and multifaceted, comprising a complex interplay of genetic, metabolic, behavioral, environmental, and socioeconomic factors, among others [7,8]. The World cancer research fund attributes obesity to two broad causes: lack of physical activity and poor diets characterized by high consumption of processed foods (foods that have been altered from their natural state through various methods to enhance shelf life, flavor, texture, or convenience) and red meat and very low intake of fruits and vegetables, whole grains, and fiber [9]. Poor diets are also inextricably linked to the declining health of the planet by way of degradation of natural resources, greenhouse gas emissions, and loss of biodiversity [10,11]. There is growing consensus that the standard American diet, which is high in processed foods, refined carbohydrates and added sugars, unhealthy fats, high fat dairy, and red meat, while low in fruits, vegetables, nuts and seeds, and whole grains, is unsustainable for promoting long-term human and planetary health [12,13].

Sustainable diets are defined by the Food and Agriculture Organization (FAO) as diets that promote all dimensions of an individual’s health and wellbeing, have low environmental pressure and impact, are accessible, affordable, safe, and equitable, and are culturally acceptable [14]. Plant-based diets (PBDs) are generally defined as diets that maximize the consumption of nutrient-dense whole plant foods and minimize the intake of processed foods, oils, and animal foods (including high-fat dairy products and eggs). In essence, they emphasize the intake of whole grains, fruits, vegetables, seeds, nuts, beans, lentils, soybeans, and herbs and spices. PBDs support nutrition security and human health and are associated with reduced risk of most of the top ten leading causes of death in America [15,16,17,18,19,20]. They are similarly associated with reduced greenhouse gas emissions and environmental degradation, have low environmental pressure and impact, and consequently promote environmental and ecological health [21,22,23,24,25]. PBDs are also associated with lower body weight and a decline in weight gain [15,26,27,28,29,30].

Current research underscores the benefits of PBDs, and several scientific and regulatory bodies have consistently recommended them. The Academy of Nutrition and Dietetics (the world’s largest organization of food and nutrition professionals) has issued a position paper on vegetarian diets. They have been commended as being healthful and nutritionally adequate for all stages of the lifecycle, including pregnancy, lactation, infancy, childhood, adolescence, and for athletes, when appropriately planned [31,32]. The American Institute for Cancer Research (AICR) has also recommended PBDs as being protective against cancer [33]. Additionally, the 2020–2025 *Dietary Guidelines for Americans*, like the previous guidelines (2015–2020), continue to recommend and highlight the benefits of plant-based eating patterns [34].

Despite the potential benefits of plant-based nutrition, its adoption in the USA remains low, with less than 10% of Americans following a PBD [35,36,37]. Encouraging the US population to transition from the standard American diet to adopt more minimally processed PBDs requires the involvement of various public health stakeholders who are the gatekeepers of nutritional education. Dietitians and nutritionists are specifically trained in the application of food, nutrition, and dietetics to promote public health and well-being. However, studies show that dietetic practitioners have knowledge gaps and low self-efficacy regarding plant-based nutrition and are less likely to recommend PBDs to clients. A study performed by Lea and associates [38] reported that although patients were willing to try PBDs in Europe, health care providers were less likely to recommend lifestyle modification (including adopting PBDs) as a form of disease management. They cited patients’ unwillingness to adopt PBDs and lack of adequate information about plant-based nutrition as their reasons for not recommending plant-based nutrition to their clients. Contrarily, a study performed by Morton and colleagues [39] revealed that 55% of patients were more willing to implement a PBD for three weeks if a nutritionist or dietitian recommended it.

This narrative review aims to explore and synthesize existing research on attitudes and perceptions of health professionals (including dietitians, nutritionists, physicians, nurses, and other health care professionals) towards PBDs. It also examines barriers and enablers influencing their incorporation of PBDs in routine patient care.

### 1.1. Plant-Based Diets

Plant-based diets generally emphasize the consumption of whole grains, fruits, vegetables, nuts, legumes, and seeds while minimizing the consumption of processed foods, oils, and animal products [40,41]. There are several variations of PBDs, including: lacto-vegetarian and ovo-vegetarian diets, lacto-ovo vegetarian diets, pesco-vegetarian diets, and vegan diets [11,42]. There are other diets related to PBDs, such as the Mediterranean diet and the Dietary Approaches to Stop Hypertension (DASH) diet, which call for the reduced consumption of meat and animal products [22,43,44]. Table 1 below provides a breakdown of common PBDs.

### 1.2. Benefits of Plant-Based Diets

Diet- and lifestyle-related chronic diseases are the leading causes of death in the developed world and in the U.S. [25,45]. However, in most cases, these diseases can be prevented through lifestyle and dietary changes [46]. Diets low in sugar, sodium, refined grains, processed foods, and animal-based foods can substantially benefit both human and planetary health and reportedly save the global economy between 1 trillion to 31 trillion US dollars which is equivalent to between 0.4% and 13% of global gross domestic product (GDP) [47].

### 1.3. Theoretical Domains Framework (TDF)

The Theoretical Domains Framework (TDF) was developed for the implementation of research to identify influences of health professionals’ behavior in relation to the implementation of evidence-based recommendations. The TDF integrates 33 theories of behavior and behavioral change and 128 key theoretical constructs related to behavioral change into a single framework with 14 theoretical domains, which cover the main factors influencing practitioner clinical behavior and behavioral change. The 14 domains include: knowledge (knowledge about a condition or scientific rationale), skills (competence/skill assessment), social/ professional role and identity (influence of societal and professional roles on behavior), beliefs about capabilities (self-efficacy/self-confidence to perform behavior), optimism, beliefs about consequences/anticipated outcomes/attitude, reinforcement, intentions, goals, memory, attention and decision processes, environmental context and resources (environmental constraints), social influences, emotion, and behavioral regulation [48,49,50,51].

## 2. Materials and Methods

The methodology outlined by Arskey and O’Malley [52] was used, which includes identifying the research question, finding relevant research studies, study selection, compiling data, and summarizing and reporting findings.

### 2.1. Research Question

This review seeks to explore an important research question: what factors serve as enablers and barriers for health care professionals regarding their recommendation of plant-based nutrition to their patients? By examining the existing literature, this review will investigate the influences that encourage or hinder health care professionals in promoting plant-based nutrition as part of patient care.

### 2.2. Literature Search

The literature search was conducted using the PubMed and Web of Science databases, selected for their extensive coverage of biomedical and clinical research, as well as their robust citation metrics and comprehensive indexing. These databases were accessed through Kansas State University. The search was performed on 18 April 2024. To investigate attitudes and perceptions of health professionals, the search terms used included the following: [Attitude OR perception OR view OR opinion OR belief OR “Attitude of Health Personnel”] AND [Dietitian OR nutritionist] OR health professional OR Health Personnel] AND [Vegan OR vegetarian OR plant-based diet OR Mediterranean diet] OR DASH diet] OR “dietary approaches to stop hypertension Diet” OR “Plant-Based”]. All research articles published in English with at least one search term from each category were considered, and no filter was applied regarding year of publication.

### 2.3. Eligibility Criteria

Peer-reviewed studies in the English language available in full text were included. The articles included had to be investigating views, opinions, attitudes, or perceptions of health professionals towards any of the plant-based diets/terms used in the search terms. In case of interventions, the evaluation/assessment had to have been performed prior to the intervention. Elimination criteria included the following: articles not written in English, commentaries or reviews, studies where only health professionals following a vegan or plant-based diet were involved, studies where intervention preceded assessment, and studies where subjects were not health professionals.

### 2.4. Data Profiling and Synthesis of Results

An Excel table was created to extract relevant data from the 27 articles, including title, author, year of publication, country of origin, study design, key study objectives, sample size, methodology, and key findings. An inductive qualitative analysis approach was employed using an iterative process of coding and comparison across the 27 studies to identify key themes emerging from the articles, which were related to health professionals’ attitudes, perceptions, barriers, enablers, and behavioral practices in relation to PBDs. Data from each study was coded and analyzed in Microsoft Excel (2021, Microsoft Corp, Redmond, WA, USA) for themes using thematic analysis [53]. Descriptive codes that captured key concepts related to the research objectives were created, and these initial codes were then refined and grouped into preliminary factors/themes. A cross-study comparison was conducted to validate and refine these preliminary factors/themes. Codes from individual studies were cross-referenced with those from other studies to identify consistent, overarching patterns.

This iterative process allowed for the synthesis of comprehensive themes reflecting the breadth and depth of findings across the 27 studies. Following the inductive analysis, a deductive approach was used to map and align identified themes with the domains of the Theoretical Domains Framework. Each theme was reviewed and aligned with a relevant TDF domain to stratify barriers and enablers influencing health professionals’ attitudes and behavior towards implementing PBDs in routine care for patients.

### 2.5. Article Search and Selection

The initial search yielded 151 articles. This was narrowed down to 31 articles after title and abstract screening and elimination of 17 duplicate articles. After full article screening, eight articles were further excluded because they did not meet the eligibility criteria. Cross-referencing Artificial Intelligence (AI) tools, Scispace and Bunni yielded four more articles, which brought the total number of articles used for the review to twenty-seven [54,55]. The flow of the article selection process is depicted in Figure 1.

## 3. Results

### 3.1. General Overview of Included Studies

All of the articles were published between 2015 and 2024, except for one article, which was published in 1999. The majority [62.9%] were published between 2020 and 2024. Twenty-three of the twenty-seven articles used questionnaires for data collection. Two used interviews exclusively, and the other two used both interviews and questionnaires. The articles came from 12 countries, with most (15/27) articles coming from the USA (9/27) and Australia (6/27). Other countries were Canada (2/27), the UK (2/27), Spain (2/27), France (1/27), Peru (1/27), New Zealand (1/27), Israel (1/27), South Africa (1/27), Italy (1/27), and the Netherlands (1/27). Nine studies were conducted with only dietitians or nutritionists [56,57,58,59,60,61,62,63,64], eight included dietitians/nutritionists and various health professionals, and ten studies were conducted with health professionals not including dietitians or nutritionists. Details of characteristics pertaining to the articles are depicted in Table 2.

### 3.2. General Overview of Health Professionals’ Attitudes and Perceptions

Currently, there is a paucity of research investigating attitudes and perceptions of health professionals towards plant-based nutrition. PBDs were perceived favorably by health professionals except in three studies [69,71,74]. Reasons for these negative attitudes include the association between PBDs and a higher risk of nutrient deficiencies among pregnant women and among children [69,74] and health professionals linking PBDs with eating disorders and consequently nutrient deficiencies among youth with eating disorders [71]. Positive attributes associated with PBDs included being healthy [72,77], management of chronic conditions such as chronic kidney disease, type 2 diabetes, and cardiovascular disease (CVD), among others [56,58,60,65], reducing the risk of chronic co-morbidities, weight loss [68], being more environmentally sustainable [79] and others.

### 3.3. Factors Influencing Health Professionals’ Attitudes and Perceptions Towards Plant-Based Diets

Nine key themes were identified during the analysis as determinants influencing health professionals’ perceptions (encompassing attitudes, viewpoints, and opinions) and behavioral practices regarding plant-based nutrition. In this study, perceptions refer to the cognitive and affective factors, including attitudes (feelings towards PBDs), viewpoints (broader professional perspectives shaped by experience), and opinions (specific beliefs). Behavioral practices refer to clinical actions and routine behaviors related to recommending and prescribing PBDs in patient care. Identified themes were knowledge, education and training, evidence-based guidelines, multidisciplinary collaboration, personal experience and interest, educational resources for both patients and health professionals, lack of time, safety and compliance challenges, and lack of confidence in patient capabilities. These themes were mapped with TDF domains (based on theoretical relevance and empirical evidence) to stratify enablers and barriers to the implementation of PBDs in routine care for patients, as shown in Table 3. The most salient TDF domains determined to be strongly linked to these themes were environmental context and resources (n = 5), skills (n = 4), social/professional role and identity (n = 3), beliefs about consequences (n = 3), and knowledge (n = 2), where n refers to the number of key themes coded to a particular domain. Other domains that were identified were optimism, goals, emotion, beliefs about capabilities, and social influences.

#### 3.3.1. Knowledge

Knowledge was identified in 12 of the 27 studies that were analyzed as a key factor that can either enable or act as a barrier for health professionals implementing and recommending plant-based nutrition. Generally, the majority of health professionals considered their knowledge about plant-based nutrition insufficient and inadequate [57,67,68,69]. Most lacked knowledge about the definitions of PBDs, the key principles behind them, their benefits for human health, disease management, and planetary health, as well as the robust scientific evidence supporting their application through various stages of the lifecycle, among others. Health professionals with a history of following PBDs were found to be more knowledgeable about PBDs than their counterparts who had never tried any version of PBDs. High knowledge scores in some studies were found to be positively correlated with positive attitudes towards PBDs [74].

#### 3.3.2. Education and Training

Ten of the twenty-seven studies reviewed reported that participants indicated that their university and professional education and training had not equipped them with the education and skills related to plant-based nutrition and therefore felt less confident about discussing and implementing it in their practice [60,62,68,69,70,73]. Harkin and colleagues [81] reported that in a sample of 140 physicians and 96 cardiologists, only 13.5% agreed that their academic training had prepared them to discuss nutrition with their patients. Within this group, 78.4% thought that additional training in nutrition would help them provide better clinical care in the prevention of cardiovascular diseases. For many, the scientific literature was not the main source of information but rather the media, online sources, and social settings. Several studies reported education and training as enablers for health professionals to discuss and recommend plant-based nutrition with their patients [37,56,64,65,70,71].

#### 3.3.3. Evidence-Based Guidelines

A few studies suggested that some health professionals were aware of the scientific research supporting claims about the benefits of plant-based nutrition, particularly in the prevention and management of chronic diseases [37,58,60]. In other studies, participants were not aware of the scientific evidence backing claims made about the benefits of plant-based nutrition, and as such, these health professionals were more reluctant to discuss or recommend plant-based nutrition in their practice [60,68,70]. Some health professionals indicated that having robust evidence-based guidelines/summaries of research findings regarding plant-based nutrition would increase their self-efficacy and enable them to discuss and recommend it to their clients [37,58,60,65,70]. One of the participants in Lee and colleagues’ [37] study is quoted as saying that, “there is a lack of clear clinical practice guidelines and diet-specific educational support.”

#### 3.3.4. Multidisciplinary Collaboration

In connection with evidence-based guidelines, a few studies also highlighted the need for collaborations across various health/scientific disciplines involved in providing health care services relating to diet and nutrition to patients. They opined that having consistent messaging would avoid causing confusion to clients [56,58,68,70]

#### 3.3.5. Personal Experience and Interest

Health professionals were more inclined to provide regular counseling on plant-based nutrition if they personally adhered to it most of the time or always, in contrast to only occasionally or less frequently [60,77]. Studies showed that most participants did not adhere to or have any personal experience with any of the PBDs [37,64,76,77,80]. Consequently, they perceived them as unrealistic, complicated, difficult to sustain, lacking in variety, not fulfilling, and cost-prohibitive, among other reasons. On the other hand, participants who had tried some of the PBDs were found to have more positive attitudes. These were driven by factors such as curiosity, environmental and ethical concerns, health benefits, and factors related to personal preference regarding taste, cost, and ingredients [62,77,80].

#### 3.3.6. Educational Resources

Several participants indicated that access to opportunities for practical-based professional development, such as scientific conferences, continuous training programs, and plant-based nutrition-related education resources [60] and tools for both health professionals and their clients, would enhance their ability to deliver improved clinical care to their patients. These would have to be evidence-based, easily accessible, and visually appealing and could be in the form of mobile phone applications with clear and concise messaging, handouts, posters, recipes, cookbooks, menu plans, and food swaps, among others [56,58,60,79].

#### 3.3.7. Lack of Time

Study participants also reported that they were limited by time constraints regarding keeping up with the literature and had limited clinician time to discuss and counsel patients on plant-based nutrition [59,60,65,68,75,76,78]. Time constraints were also linked to the inability of patients to adopt PBDs with respect to food preparation because patients tend to prioritize convenience over other factors.

#### 3.3.8. Safety and Compliance Challenges

Study participants expressed fear around potassium control in patients with chronic kidney disease, especially in instances of comorbid conditions such as diabetes and CVD. There were concerns about prescribing dried fruit, nuts, and seeds regarding potassium control and fear of inducing hyperkalemia and/or hyperglycemia [56,58,65,75].

#### 3.3.9. Lack of Confidence in Patient Capabilities

Several health professionals expressed a lack of belief in patients’ capabilities to change behavior and improve diet adherence. Some opined that PBDs were “not realistic for the patient,” and that patients are not interested in plant-based nutrition and have a knowledge deficit regarding the diet–disease relationship. They also asserted socioeconomic challenges, culturally diverse backgrounds coupled with long-held unhealthy eating patterns, and heavy reliance on convenience foods as key challenges to aligning diet education regarding patient adoption of plant-based nutrition [65,70]. In contrast, some expressed support for single-nutrient-based advice as more straightforward, with evidence of clearer links to management of specific clinical risk markers [70]. Some patients were reported as unwilling to have appointments with a dietitian [68]. Participants working in private versus public settings were also more likely to strongly agree that they were confident to counsel patients on plant-based nutrition [68].

## 4. Subgroup Analysis of Barriers and Enablers by Professional Category

The studies included in this analysis explored the knowledge, attitudes, and perceptions of various health professionals regarding PBDs; however, most did not account for the distinct roles, responsibilities, and levels of nutrition training across the various professional categories. In response to differences in nutrition training and clinical responsibilities among health professionals, a subgroup analysis was conducted to identify key barriers and enablers related to PBD recommendations by professional category (physicians/clinicians versus dietitians/nutritionists) (Table 4).

## 5. Discussion and Conclusions

This review assessed health professionals’ attitudes and perceptions and the key barriers and enablers influencing their integration of PBDs into routine patient care, with the overarching goal of informing strategies to promote wider adoption of plant-based nutrition among Americans. The review revealed that health professionals often felt unprepared and uncertain about including plant-based nutrition in their daily practice. This study identified nine themes that influenced health professionals’ attitudes and behaviors/practices regarding plant-based nutrition. By aligning these themes with the domains of the Theoretical Domains Framework (TDF), the study highlights important enablers and facilitators that could promote behavior change among health professionals, while also highlighting barriers to the integration of plant-based nutrition into routine patient counseling.

The most prominent domain highlighted by this study was environmental context and resources. This reveals that whether a health professional’s environment or circumstances support or hinder the application of plant-based nutrition in daily practice is a key factor. A systematic review by Boocock and colleagues [82] examining clinicians’ perceived barriers and enablers to dietary management of adults with type 2 diabetes also reported environmental context and resources as the most significant TDF domain in their study. Similarly, a study by Mayr and associates [68] exploring clinician perspectives of barriers and enablers to implementing the Mediterranean dietary pattern in routine care for both coronary heart disease and type 2 diabetes also reported environmental context and resources to be the dominant domain. In the present study, this domain encompassed education and training, multidisciplinary collaboration, educational resources, and lack of time. Misinformation was also identified as a barrier. Limited patient consultation time and educational resources have been recognized in other studies as barriers to health professionals providing nutritional education and integrating evidence-based practices such as plant-based nutrition in routine care [83,84].

“Skills” was the second most prominent TDF domain. It was related to four of the nine identified influencers of health professionals’ behaviors in relation to plant-based nutrition. This domain, according to Atkins and colleagues [51], relates to proficiency acquired through practice and encompasses skills development, competence, ability, interpersonal skills, practice, and skills’ assessment as constructs. Lack of or limited skills was identified in several studies as a barrier to health professionals discussing and recommending plant-based nutrition to patients [60,70]. Additionally, lack of skills such as meal planning and cooking skills was also identified as a barrier for patients’ adoption of PBDs [37,56,60,75]. In the current study, this domain encompassed themes such as knowledge, education and training, and evidence-based guidelines. Access to relevant evidence-based research summaries and guidelines was considered an enabler; however, the lack of time to keep up to date with the relevant scientific literature was a barrier. Other studies have also identified time constraints for finding and reviewing the scientific information, limited skills in critically analyzing the scientific literature, and a lack of research applicable to everyday practice as key barriers to incorporating plant-based nutrition in routine care [85,86]. Research shows that improved access to skill-based professional training on PBDs, coupled with consistent integration into university curricula, would enhance health professionals’ knowledge, skills, confidence, and self-efficacy in delivering evidence-based nutritional education. Furthermore, these opportunities could be more effectively supported and integrated within existing health care frameworks to further strengthen outcomes [82,84,87,88,89,90].

Other significant domains were social/professional role and identity and beliefs about consequences. The social/professional role and identity domain encompasses aspects like professional identity, role, social identity, professional boundaries, professional confidence, group identity, and leadership. Several studies revealed that participants reported inadequate professional training as a barrier to discussing and recommending PBDs to clients. In contrast, professional development opportunities such as scientific conferences, ongoing training programs, and PBD-based educational resources were seen as facilitators for discussing and recommending PBDs during routine practice. A lack of multidisciplinary collaboration where knowledge is exchanged both within and across disciplines was identified as a barrier and a major source of misinformation when providing nutritional education to clients [37,56,64,65,68,71]. Health professionals who were well-informed about evidence-based research and current dietary guidelines were more likely to recommend plant-based nutrition to their clients. This further emphasizes the importance of improved access to practice-focused professional development on plant-based nutrition [68,83].

The domain of beliefs about consequences encompasses expectations about outcomes, characteristics of those outcomes, and consequences [49,51]. Boocock and colleagues [82] suggest that health professionals’ beliefs regarding the consequences of interventions, such as recommending plant-based nutrition in the management of chronic conditions, may lead to reservations about their effectiveness for patients. Mayr and colleagues [68] found that clinicians’ lack of optimism and belief in the potential consequences led them to doubt that recommending a Mediterranean dietary pattern (a type of PBD) would improve clinical outcomes for patients with coronary heart disease and type 2 diabetes. This perspective contrasts with the positive findings from an umbrella review of meta-analyses by Dinu and colleagues [91], which included 13 meta-analyses of observational studies and 16 meta-analyses of randomized controlled trials. The review covered a total of 12.8 million subjects and investigated 37 health outcomes, including cardiovascular outcomes, cancer outcomes, cognitive disorders, and metabolic disorders, among others.

Subgroup analysis by professional category (physicians/clinicians versus dietitians/nutritionists) highlighted unique barriers and enablers that both groups face in relation to recommending PBDs. Physicians often cited time constraints, limited nutritional education during medical and professional training, and uncertainty regarding the evidence base as key barriers, whereas dietitians and nutritionists more frequently highlighted limited resources and patient resistance. These findings underpin the importance of tailored strategies to address profession-specific needs. For physicians/clinicians, enhancing undergraduate and continuing medical education in nutrition, more specifically, evidence-based guidance on PBDs, could increase their confidence in prescribing PBDs. For dietitians and nutritionists, efforts may be better focused on improving institutional support, resources, and interdisciplinary collaboration to enhance long-term dietary counseling.

Credible scientific studies demonstrate that adopting a PBD is linked to better health outcomes, reduced body weight, and reduced long-term weight gain, positioning it as a promising strategy in combating the obesity epidemic [39,92,93,94,95,96]. Additionally, PBDs support the preservation of biodiversity and planetary health [11,97,98,99,100] while also being affordable and culturally acceptable [100]. Recent updates by a limited number of professional societies have begun to reflect the growing evidence base supporting PBDs; however, these updates remain the exception rather than the norm. Consequently, it is understandable that many health professionals may lack adequate training or confidence in recommending PBDs in clinical practice. While the U.S. Dietary Guidelines for 2020–2025 encourage the inclusion of more plant-based foods, they do not fully align with more progressive international frameworks such as the EAT-Lancet commission’s planetary health diet or the World Health Organization’s recommendations on sustainable healthy diets, which integrate both human health and environmental sustainability [10,34]. This disparity highlights the need to update national and institutional guidelines to reflect current scientific consensus on the benefits of plant-based nutrition.

To increase the adoption of PBDs within the American population, it is crucial to prioritize strategies that support health professionals in counseling on plant-based nutrition while also addressing identified barriers. Health professionals need continuous opportunities to enhance their knowledge and skills through targeted training programs, workshops, and conferences related to PBDs. Multidisciplinary collaboration between dietitians, physicians, nurses, and other health professionals is also crucial for sharing expertise and providing comprehensive care. Access to up-to-date, evidence-based guidelines and resources will further empower health professionals to confidently recommend plant-based nutrition, ensuring that patient care is informed by the latest scientific research [101]. For instance, the American College of Lifestyle Medicine (ACLM) provides comprehensive evidence-based clinical practice guidelines and toolkits designed to support the prescription of whole-food plant-based diets for chronic disease prevention and management [102]. Kaiser Permanente, one of the largest not-for-profit health care providers in the U.S., has published clinical guidelines promoting PBDs for the prevention and treatment of chronic diseases [103,104]. In addition, online platforms such as Nutritionfacts.org, curated by Dr. Michael Greger, and the Physicians Committee for Responsible Medicine offer regularly updated educational materials and clinical support tools for both health professionals and patients [105].

Policymakers should draft policies and update food-based dietary guidelines to align with the current scientific literature and prioritize public health concerns such as reduced risk of chronic diseases and premature mortality. This information should be made available to all health professionals involved in nutrition counseling, including physicians, nurses, dietitians, nutritionists, extension agents, and others. Policies and guidelines should also be clearly communicated and widely disseminated to the public through popular media channels, ensuring clarity and consistency to avoid confusion and conflicting messages. Since a lack of personal experience with plant-based nutrition principles hinders the ability to recommend them to patients, experiential education, such as culinary nutrition, could be an effective strategy to enhance health professionals’ knowledge of plant-based nutrition and boost their self-efficacy, while culinary programs in schools and community centers for both adults and children could also help develop practical cooking skills for PBDs [106,107,108,109,110,111].

This study provides valuable insights into health professionals’ attitudes and perceptions towards plant-based nutrition and barriers and enablers to its prescription; however, it is not without limitations. The predominance of observational studies in the analysis limits the generalizability of the findings. Although the inclusion of studies from multiple countries and a variety of health professionals provides a more diverse perspective, the overall evidence remains limited by study design. To overcome these limitations, future studies should aim to incorporate a greater number of studies, including longitudinal studies, to enhance the robustness and applicability of results.

## Figures and Tables

**Figure 1 nutrients-17-02095-f001:**
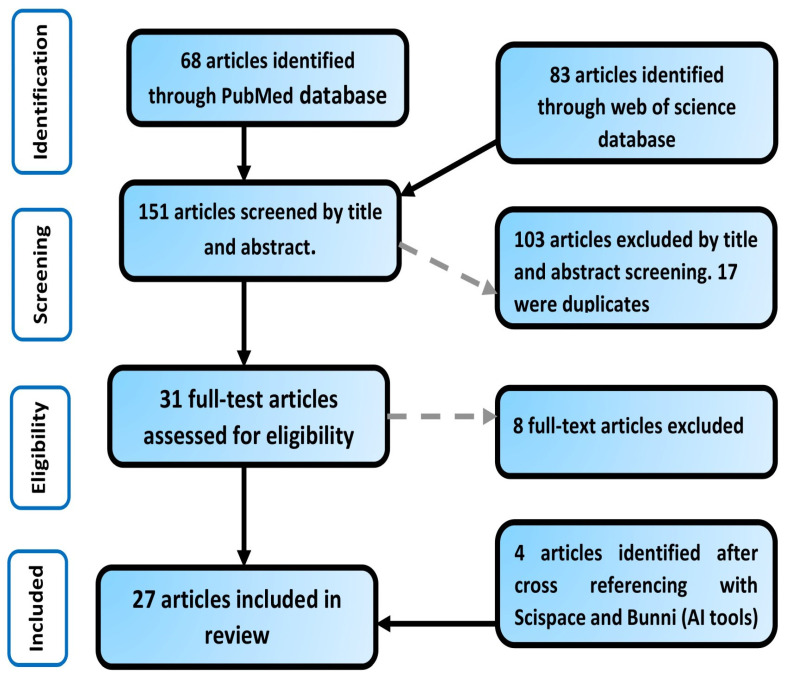
Flow diagram of review selection process from initial search to final number of included studies.

**Table 1 nutrients-17-02095-t001:** Dietary Patterns That Emphasize the Consumption of Plant Foods.

Dietary Pattern	Foods
Lacto-vegetarian diet	Includes dairy
Ovo-vegetarian diet	Includes eggs
Lacto-ovo vegetarian diet	Includes dairy and eggs
Pesco-vegetarian diet	Includes fish and seafood
Vegan diet	Excludes all meat and all animal products
Mediterranean diet	Based on fruits, vegetables, whole grains, legumes, moderate consumption of dairy and fish, and low consumption of meat and sweets
DASH diet	Based on vegetables, fruits, and whole grains; includes fat-free and low-fat dairy products, fish, poultry, beans, and nuts.

**Table 2 nutrients-17-02095-t002:** Summary of articles included in the review with key characteristics.

Author, Year	Country	Study Design	Population and Sample	Objective	Methodology	Key Findings
Stanford et al., 2022 [56]	Australia	Cross-sectional	N = 35 renal dietitians completed online surveys, and 11 participated in in-depth interviews	Explore perspectives of renal dietitians regarding PBDs for chronic kidney disease (CKD) management and evaluate their acceptability of a hypothetical plant-based dietary prescription	Exploratory Mixed methods: Short online questionnaire and in-depth semi-structured interview	Renal dietitians perceived PBDs as beneficial to patients with CKD.
Betz et al., 2022 [65]	USA	Cross-sectional	N = 382 dietitians (154 physicians, 62 nurse practitioners, 32 fellows, 13 physician assistants, and 14 other professionals)	Understand the perspectives of nephrology professionals towards the use of PBDs for the treatment of CKD	Online questionnaire based on a previous survey	Nephrology professionals believed PBDs were beneficial in the management of CKD, but dietitians were more likely to be aware of the benefits of PBDs than other professionals.
Fuller & Hill, 2022 [66]	UK	Cross-sectional	N = 116 specialist eating disorder professionals, 90 general mental health professionals, and 186 other professionals	Investigate attitudes of health care professionals towards veganism	Self-reported questionnaire based on general eating habits and ATvegan questionnaires	All had positive views of veganism, but general mental health professionals had more positive attitudes than eating disorder specialists and other professionals.
Bettinelli et al., 2019 [67]	Italy	Cross-sectional	140 nurses,135 pediatric nurses, 60 midwives, 43 health care support workers, and 40 staff nurses	Assess knowledge of health care professionals regarding the adoption of vegetarian diets from pregnancy through adolescence	Online questionnaire developed for the study and pre-tested	Clinicians had a positive view of the Mediterranean diet (MD), though it was not routinely recommended due to limited knowledge, practice skills, and training
Hughes et al., 2014 [57]	USA	Cross-sectional	N = 136 dietitians, of whom 124 were registered dietitians	Assess dietitians’ perceptions of plant-based protein quality	Online questionnaire developed for the study and pre-tested	Dietitians had a positive attitude towards PBDs, but knowledge about plant-based protein quality was limited.
Moutou et al., 2021 [58]	UK	Cross-sectional	N = 12 registered dietitians	Explore dietitians’ views about advising on 5 dietary patterns (including MD and DASH diets) deemed effective for the management of type 2 diabetes	Semi-structured interviews with short demographic questionnaires developed for the study	Study participants considered the MD effective, but most had mixed responses about the DASH diet.
Mayr et al., 2022 [68]	Australia	Cross-sectional	N = 57 clinicians (21 nurses, 19 doctors, 13 dietitians, and 4 physiotherapists)	Explore multidisciplinary health care professionals’ perspectives on recommending the MD to patients with coronary heart disease and type 2 diabetes	Qualitative study with individual semi-structured interviews via telephone or face-to-face	The MD was not routinely recommended and clinicians had limited knowledge and practice skills regarding MD; barriers to recommending the MD were a lack of education and training and personal experience/interest.
Meulenbroeks et al., 2021 [69]	The Netherlands	Cross-sectional	N = 411 (121 midwives, 179 obstetricians, and 111 dietitians)	Evaluate self-reported knowledge and advice given by Dutch obstetric caregivers and dietitians to pregnant women following PBDs	Online questionnaire developed based on focus group interviews	Both obstetricians and midwives reported limited knowledge about strict PBDs. Only 38.7% of dietitians felt they had enough knowledge to advise pregnant women on strict PBDs. They believed that women following a strict PBD during pregnancy were at a higher risk of nutrient deficiencies.
Mayr et al., 2022 [70]	Australia	Cross-sectional	N = 14 (7 doctors, 3 nurses, 3 dietitians, and 1 exercise physiologist)	Assess multidisciplinary clinicians’ perspectives on whether the Mediterranean diet (MD) is recommended in routine management of non-alcoholic liver disease	Semi-structured individual phone and face-to-face interviews	The MD was seen as an evidence-based approach for enhancing diet quality, promoting weight loss, and reducing the risk of chronic co-morbidities. However, some doctors and nurses had limited knowledge of the specific literature supporting the benefits of following an MD.
Hawkins et al., 2019 [59]	USA	Cross- sectional	N = 205 nutrition and dietetics program directors	Investigate curricular practices in accredited dietetics programs and assess the prevalence and perceived importance of vegetarian and vegan nutrition instruction	Online questionnaire developed for the study and pre-tested	Over 90% of program directors agreed that vegetarian nutrition should be taught, while 87% agreed that vegan nutrition should be taught. Program directors in northeastern programs had higher percentages of agreement than those in southern programs. In addition, 51% and 49% of the programs teach vegetarian and vegan nutrition, respectively.
Albertelli et al., 2024 [71]	France	Cross- sectional	N = 18 (14 dietitians, 3 physicians specialized in nutrition, and 1 psychiatrist)	Investigate health care professionals’ subjective experience of vegetarianism in patients with eating disorders (EDs)	Qualitative study with remotely administered semi-structured interviews via videoconferences and telephone	Health professionals regarded vegetarianism as a restrictive approach and often linked it to eating disorders in patients. They were strongly opposed to veganism, citing the risk of severe nutritional deficiencies.
Mayr et al., 2020 [60]	Australia	Cross- sectional	N = 182 dietitians who had practiced with at least one of the relevant chronic disease patient groups	Evaluate the extent the MD is routinely recommended by dietitians to patients with chronic diseases	Online questionnaire based on TDF	Approximately 62%, 46%, and 39% of dietitians strongly agreed that there was enough evidence to support recommending MD to patients with CVD, type 2 diabetes, and non-alcoholic liver disease, respectively. Moreover, 48% strongly agreed that they were knowledgeable about the principles of MD, and 46% were confident in counseling patients about MD.
McHugh et al., 2019 [72]	New Zealand	Cross- sectional	N = 41 (20 doctors, 13 nurses, 7 pharmacists, and 1 osteopath)	Investigate whether health professionals have sufficient nutritional education for their roles in health education and promotion and whether their nutritional beliefs are consistent with the current literature	Mixed methods, including an online de novo questionnaire and one focus group	PBDs were generally viewed as beneficial to health but deemed complicated. Moreover, 43% of participants reported dissatisfaction with the amount of nutritional training received.
Olfert et al., 2020 [73]	USA	Descriptive case study	N = 29 health professionals, 15 currently practicing in cohort 1 and 14 aspiring health professionals in cohort 2 from various disciplines	Determine the effectiveness of culinary medicine and MD to enhance the nutritional knowledge, attitudes, and self-efficacy of current and aspiring (student) health professionals	Online questionnaire developed but influenced by evidence-based sources	At baseline, cohort 2 had higher attitude and knowledge scores. There was no significant difference in mean self-efficacy scores or mean MD adherence scores.
Hamiel et al., 2020 [74]	Israel	Cross- sectional	N = 270 pediatricians, 14.1% were following a vegetarian diet	Assess the knowledge and attitudes of pediatricians towards vegetarian diets	Online questionnaire based on a previously validated questionnaire	Pediatricians had knowledge gaps regarding vegetarian nutrition, and most did not have a positive attitude towards vegetarian diets. Knowledge was positively correlated with attitude.
Lessem et al., 2020 [75]	USA	Experiential education program	N = 30 (13 nurse practitioners, 14 registered nurses, and 3 physicians)	Increase knowledge and acceptance of whole-food plant-based (WFPB) diets and the likelihood of counseling patients about the diet among health care workers	Online questionnaires based on previously validated research	Pre-intervention average knowledge scores were 65.4%. Average self-efficacy scores for knowledge and counseling were 2.64 and 2.38 at baseline on a scale of 1 to 4.
Sentenach et al., 2019 [76]	Spain	Cross- sectional	N = 422 physicians (PREDIMED screener) and N = 212 physicians (knowledge/opinion survey)	Evaluate physicians’ knowledge/awareness of and adherence to an MD	Online questionnaire based on the PREDIMED MD screener previously used in the PREDIMED study	Most physicians did not adhere to the MD, but 70% considered themselves knowledgeable about the benefits of the MD, and 60% were willing to recommend it to patients.
Estell & Hughes, 2021 [77]	Australia	Cross- sectional	N = 660 (228 nutrition professionals)	Explore consumer and nutrition professional perceptions and attitudes to plant protein, including plant-based meat alternatives	Online questionnaire based on previous research	Over 80% of nutrition professionals agreed that following a PBD promoted good nutrition, and over 70% disagreed that it was hard to meet protein requirements while following a PBD.
Asher et al., 2021 [61]	Canada	Cross- sectional	N = 403 dietitians	Assess Canadian registered dietitians’ attitudes and behaviors towards the new food guidelines’ increased plant-based recommendations	The online questionnaire developed for the study and pre-tested	Over 80% of dietitians considered the food guide’s recommendation to choose plant-based protein foods as evidence-based. Most had a positive view of the new guidelines, and 58.7% were more likely to encourage their clients to select plant-based protein options.
Aggarwal et al., 2019 [78]	USA	Cross- sectional	N = 303 physicians from departments of cardiology and general medicine	Assess nutrition and exercise knowledge and personal health behaviors of physicians	Online questionnaire based on validated surveys	Less than 25% of the physicians in the study followed the facets of MD.
Saintila et al., 2021 [62]	Peru	Cross- Sectional	N = 179 registered dietitians (72 vegetarians and 107 non-vegetarians)	Compare the level of knowledge of vegetarian and non-vegetarian Peruvian dietitians regarding vegetarianism	Online questionnaire based on the recommendations of the current dietary guidelines	Vegetarian dietitians were more knowledgeable about the risks and benefits associated with vegetarian diets.
Janse et al., 2021 [63]	South Africa	Cross- Sectional	N = 101 dietitians (45 government employed and 48 in private practice)	Assess whether dietitians in South Africa would use a whole-foods plant-based diet (WFPBD) to address chronic diseases	Online questionnaire based on validated surveys	A significant number of dietitians reported inadequate university training surrounding PBDs, albeit a significant number of them were confident about prescribing PBDs to clients.
Duncan & Bergman, 1999 [64]	USA	Cross- sectional	N = 183 registered dietitians from Vermont, Nebraska, and Washington	Investigate what registered dietitians know about the safety, adequacy, and health benefits of vegetarian diets	Paper questionnaire sent by mail	Average knowledge and attitude scores were greater for registered dietitians who were currently or had previously followed a vegetarian diet. Overall knowledge scores varied between states.
Fresan et al., 2023 [79]	Spain	Cross- Sectional	N = 2545 health professionals (550 dietitian-nutritionists, 1139 nurses, 427 physicians and 346 pharmacists, and 83 others)	Assess knowledge and attitudes regarding sustainable diets among health professionals in Spain	Online questionnaire developed for the study	Approximately 21.5% of respondents had not previously heard about sustainable diets, and 32.4% acknowledged their limited knowledge about the subject. Most when presented with information about sustainable diets considered it important to promote them.
Krause et al., 2019 [80]	USA	Cross- Sectional	N = 64 (12 residents,6 fellows, 46 physician attendings)	Assess medical providers’ knowledge of plant-based nutrition and their willingness to recommend it to patients	Online questionnaire developed for the study	Approximately 33% of respondents were willing to recommend PBDs, while the majority (51%) responded with maybe. Only 28% were willing to adopt PBDs, and 25% were willing to try it for 6 months or more.
Lee et al., 2015 [37]	Canada	Cross- Sectional	N = 98 patients and 25 health care providers	Assess awareness, barriers, and promoters of plant-based diet use for the management of type 2 diabetes for the development of an educational program	Two sets of questionnaires for patients and health care providers were developed for the study	Approximately 72% of health care providers reported knowledge of PBDs for the management of type 2, while the majority of patients (89%) had not heard of using PBDs to treat/manage type 2 diabetes. Less than 50% of respondents were aware of the benefits of PBDs regarding other chronic conditions.
Harkin et al., 2018 [81]	USA	Cross- Sectional	N = 236 (140 physicians and 96 cardiologists)	Assess basic nutritional knowledge, attitudes, and practices of physicians	Online questionnaire based on validated surveys	Nutrition knowledge was average, with only 13.5% feeling sufficiently trained to discuss nutrition with their patients. Physicians most commonly recommended the MD (55.1%), followed by the DASH diet (38.2%), to their patients.

**Table 3 nutrients-17-02095-t003:** Identified Themes Mapped with TDF Domains.

**Theme**	**TDF Domains**	**Enablers**	**Barriers**
Knowledge	- Knowledge - Skills	- Personal experience with PBDs - Knowledge of the diet–disease relationship - Adequate knowledge of PBDs and their benefits - Knowledge of scientific rationale for PBDs	- Limited knowledge of basic principles of PBDs to discuss with patients - Lack of knowledge about the benefits of PBDs - Limited knowledge and practice skills - Limited knowledge exchange within and across multidisciplinary teams.
Education and training	- Skills - Social/professional role and identity - Environmental context and resources	- Education about PBDs at university level and continuous professional evidence-based training, conferences, etc. - Patient knowledge about PBDs - Online nutritional education	- Lack of education or training at degree and professional levels - Misinformation from other health professionals and non-peer-reviewed sources, such as the Internet and media - Low self-efficacy to discuss PBDs with patients due to inadequate training
Evidence-based guidelines	- Skills - Social/professional role and identity - Beliefs about consequences	- Awareness of peer-reviewed evidence - Awareness of current dietary guidelines in support of PBDs - Access to position papers in support of PBDs from respectable scientific bodies	- Perceived lack of evidence-based, properly tested practice guidelines - Lack of access to evidence summaries - Disagreement with available evidence
Multi-disciplinary collaboration	- Social/professional role and identity - Environmental context and resources - Social influences	- Consistent messaging from various health professionals	- Misinformation from other health professionals - Limited knowledge exchange within and across multidisciplinary teams.
Personal experience and interest	- Skills - Beliefs about capabilities - Environmental context and resources	- Health professionals trying out PBDs, even if for a limited time, and counseling patients based on evidence and experience	- Lack of health professional/patient personal experience with PBDs - Lack of interest in trying PBDs even for a short time. - Providing counseling based on personal biases rather than evidence
Educational resources for both patients and health professionals	- Knowledge - Environmental context and resources	- Availability of educational materials such as meal plans, menu plans, food checklists, recipes, and mobile apps to teach and share with patients - Access to evidence summaries - Access to visually appealing content for patients	- Absence of patient education tools and resources/materials - Low confidence to discuss PBDs with patients - Limited/non-existent practical-based professional development - Access to clinical guidelines related to PBDs.
Lack of time	- Goals - Environmental context and resources	- Access to resources and tools to share with clients to use at home	- Limited time allocated to patients’ consultations - Limited time to keep up with peer-reviewed literature - Belief that patients prioritize convenience foods over food preparation due to limited time
Safety and compliance challenges	- Beliefs about consequences - Emotion	- Individual patient counselling - Access to evidence-based clinical guidelines - Having knowledge of PBD benefits	- Fear of inducing comorbidities like hyperkalemia and or hyperglycemia among patients with chronic kidney disease [CKD] - Fear around potassium control among patients with CKD - Deficiency concerns
Lack of confidence in patient capabilities	- Beliefs about consequences - Optimism	- Educating patients about PBD health benefits and key concepts - Individual patient counselling - Inclusion of evidence-based or endorsed patient resources and tools. - Goal setting around changing patient dietary patterns	- Diet presumed unrealistic for patients of a low socioeconomic background - PBDs perceived as incompatible with patient food culture and eating patterns - Patients deemed to have low health literacy/knowledge deficit of the diet–disease relationship - Assume patients are unwilling to try PBDs because they are hard/complicated

**Table 4 nutrients-17-02095-t004:** Subgroup Analysis of Barriers and Enablers by Professional Category.

Category	Physicians/Clinicians	Dietitians/Nutritionists
Barriers	Limited nutritional education, training, and practical skills at degree and professional levels	Patient resistance/culture
	Time constraints	Limited resources
	Lack of clear clinical practice guidelines and lack of knowledge of guidelines	Time constraints
	Uncertainty about scientific evidence or the benefits of PBDs	Lack of patient education and monitoring resources
	Limited resources	
	Perception that diet counseling is not their responsibility	
	Worry about overall health/fear of patient injury/risk of deficiency	
	Limited dietary knowledge exchange within and across multidisciplinary teams	
	Lack of diet-specific educational support	
	Low perceived patient acceptability	
	Low self-efficacy	
	Financial disincentives	
Enablers	Patient interest in PBDs	Professional development
	Evidence-based support for PBD benefits	Patient interest in PBDs
	Professional development	Personal history of following a PBD
	Institutional initiatives promoting preventive care	Knowledge sharing from expert dietitians
	Interest in lifestyle medicine	
	Beliefs about consequences	
	Personal history of following a PBD	

## Data Availability

The original contributions presented in this study are included in the article. Further inquiries can be directed to the corresponding author.

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
