# Peer review of "Decoding Health Professionals’ Attitudes and Perceptions Towards Plant-Based Nutrition: A Narrative Review"

_nutrients, 2025, doi:10.3390/nu17132095_

Round 1

Reviewer 1 Report

Comments and Suggestions for Authors

Abstract

The mapping mentioned in the abstract isn’t described in the methods section.

“These themes were mapped to the domains of the Theoretical Domains Framework (TDF) to identify key enablers and barriers to implementation of PBDs in routine care for patients.”

Methods

The initial search? When were other searches performed?

“The initial search was done on April 18, 2024.”

References needed for Scispace and Bunni.

What are vegan/plant-based health professionals?

“Elimination criteria included: articles not written in English, commentaries or reviews, studies where only vegan/plant-based health professionals were involved, studies where intervention preceded assessment, and studies where subjects were not health professionals.”

The following needs to be flushed out in further detail. Methods need to be described in enough detail so that others could replicate the study.

“A cross-study comparison was conducted to validate and refine these preliminary factors/themes. Factors identified within individual studies were cross-referenced with themes observed across multiple studies to identify overarching patterns. This process of comparison and synthesis allowed for the development of comprehensive themes/factors that captured the breadth and depth of the research question. A combined deductive/ inductive approach was used to map key factors/themes to the domains of the Theoretical Domains Framework.”

What are these descriptive terms in terms of the research methods - attitudes, viewpoints, opinions, and behavior/practice?

“Nine key themes were identified during analysis as key determinants influencing attitudes, viewpoints, opinions, and behavior/practice of health professionals regarding plant-based nutrition.”

Discussion

The following doesn’t match with the study’s research question.

“The purpose of this review was to assess health professionals' attitudes and percep- 325 tions toward plant-based nutrition, and how these influence their decision to incorporate 326 it into their routine practice.”

Research Question: This review seeks to explore an important research question: what factors serve as enablers and barriers for health care professionals regarding their recommendation of plant-based nutrition to their patients?

Examples of these should be provided along with references.

“Access to up-to-date, evidence-based guidelines and resources will further empower health professionals to confidently recommend plant-based nutrition, ensuring that patient care is informed by the latest scientific research.”

This sentence is too dense, which makes it hard to understand. 

“To further encourage the shift towards adopting PBDs, a holistic approach is required that integrates regulatory, technological, financial, and environmental factors [11, 104]. “

This needs to be rewritten. What are the limitations of this review study? What could be done in the next study to overcome the limitations?

“The predominance of observational studies in the analysis limits the generalizability of the findings. However, inclusion of studies from various countries with various health professionals provided a more diverse background. Future research could address some of these limitations by incorporating a larger number of studies.”

Author Response

Response to Reviewer 1 Comments

Thank you very much for taking the time to review our manuscript. Please find the detailed responses below and the corresponding revisions/corrections highlighted in red in the re-submitted file.

Comment 1: The mapping mentioned in the abstract isn’t described in the methods section.

“These themes were mapped to the domains of the Theoretical Domains Framework (TDF) to identify key enablers and barriers to implementation of PBDs in routine care for patients.”

Response 1: Thank you for pointing this out. I agree with this comment. Therefore, I have revised the methods section to include a detailed description of what was done. This can be found on page 4 line 161-179

Comment 2: The initial search? When were other searches performed?

“The initial search was done on April 18, 2024.”

Response 2: Agree. I have, accordingly modified the sentence to “The search was done on April 18, 2024.” This can be found on page 4 line 143.

Comment 3: References needed for Scispace and Bunni.

Response 3: references have been provided on page 5, line 185 and lines 638 and 639.

Comment 4: What are vegan/plant-based health professionals?

“Elimination criteria included: articles not written in English, commentaries or reviews, studies where only vegan/plant-based health professionals were involved, studies where intervention preceded assessment, and studies where subjects were not health professionals.”

Response 4: This has been clarified to “health professionals following a vegan or plant-based diet,” on page 4, line 157

Comment 5: The following needs to be flushed out in further detail. Methods need to be described in enough detail so that others could replicate the study.

“A cross-study comparison was conducted to validate and refine these preliminary factors/themes. Factors identified within individual studies were cross-referenced with themes observed across multiple studies to identify overarching patterns. This process of comparison and synthesis allowed for the development of comprehensive themes/factors that captured the breadth and depth of the research question. A combined deductive/ inductive approach was used to map key factors/themes to the domains of the Theoretical Domains Framework.”

Response 5: Thank you, I have revised the methods section to include a detailed description of what was done. This can be found on page 4 line 161-179

Comment 6: What are these descriptive terms in terms of the research methods - attitudes, viewpoints, opinions, and behavior/practice?

“Nine key themes were identified during analysis as key determinants influencing attitudes, viewpoints, opinions, and behavior/practice of health professionals regarding plant-based nutrition.”

Response 6: This paragraph has been revised to “Nine key themes were identified during analysis as determinants influencing health professionals’ perceptions (encompassing attitudes, viewpoints, and opinions) and behavioral practices regarding plant-based nutrition. In this study, perceptions refer to the cognitive and affective factors including attitudes (feelings towards PBDs), viewpoints (broader professional perspectives shaped by experience) and opinions (specific beliefs). Behavioral practices refer to clinical actions and routine behaviors related to recommending and prescribing PBDs in patient care.” This can be found on page 16, line 229-235.

Comment 7: The following doesn’t match with the study’s research question.

“The purpose of this review was to assess health professionals' attitudes and perceptions toward plant-based nutrition, and how these influence their decision to incorporate it into their routine practice.”

Research Question: This review seeks to explore an important research question: what factors serve as enablers and barriers for health care professionals regarding their recommendation of plant-based nutrition to their patients?

Response 7: Thank you for pointing this out. This has been revised to match the research question as follows: “This review assessed health professionals' attitudes and perceptions and the key barriers and enablers influencing their integration of PBDs into routine patient care, with the overarching goal of informing strategies to promote wider adoption of plant-based nutrition among Americans.” This can be found on page 21 line 347-349.

Comment 8: Examples of these should be provided along with references.

“Access to up-to-date, evidence-based guidelines and resources will further empower health professionals to confidently recommend plant-based nutrition, ensuring that patient care is informed by the latest scientific research.”

Response 8: Examples with references have been provided on page 23 line 456-463

Comment 9: This sentence is too dense, which makes it hard to understand. 

“To further encourage the shift towards adopting PBDs, a holistic approach is required that integrates regulatory, technological, financial, and environmental factors [11, 104]. “

Response 9: This sentence was removed from the discussion section.

Comment 10: This needs to be rewritten. What are the limitations of this review study? What could be done in the next study to overcome the limitations?

“The predominance of observational studies in the analysis limits the generalizability of the findings. However, inclusion of studies from various countries with various health professionals provided a more diverse background. Future research could address some of these limitations by incorporating a larger number of studies.”

Response 10: This paragraph has been revised to “The predominance of observational studies in the analysis limits the generalizability of the findings. Although inclusion of studies from multiple countries and a variety of health professionals provides a more diverse perspective, the overall evidence remains limited by study design. To overcome these limitations, future studies should aim to incorporate a greater number of studies, including longitudinal studies to enhance robustness and applicability of results.” 

This can be found on page 24, line 480-485.

Thank you!

Response to Reviewer 1 Comments

Thank you very much for taking the time to review our manuscript. Please find the detailed responses below and the corresponding revisions/corrections highlighted in red in the re-submitted file.

Comment 1: The mapping mentioned in the abstract isn’t described in the methods section.

“These themes were mapped to the domains of the Theoretical Domains Framework (TDF) to identify key enablers and barriers to implementation of PBDs in routine care for patients.”

Response 1: Thank you for pointing this out. I agree with this comment. Therefore, I have revised the methods section to include a detailed description of what was done. This can be found on page 4 line 161-179

Comment 2: The initial search? When were other searches performed?

“The initial search was done on April 18, 2024.”

Response 2: Agree. I have, accordingly modified the sentence to “The search was done on April 18, 2024.” This can be found on page 4 line 143.

Comment 3: References needed for Scispace and Bunni.

Response 3: references have been provided on page 5, line 185 and lines 638 and 639.

Comment 4: What are vegan/plant-based health professionals?

“Elimination criteria included: articles not written in English, commentaries or reviews, studies where only vegan/plant-based health professionals were involved, studies where intervention preceded assessment, and studies where subjects were not health professionals.”

Response 4: This has been clarified to “health professionals following a vegan or plant-based diet,” on page 4, line 157

Comment 5: The following needs to be flushed out in further detail. Methods need to be described in enough detail so that others could replicate the study.

“A cross-study comparison was conducted to validate and refine these preliminary factors/themes. Factors identified within individual studies were cross-referenced with themes observed across multiple studies to identify overarching patterns. This process of comparison and synthesis allowed for the development of comprehensive themes/factors that captured the breadth and depth of the research question. A combined deductive/ inductive approach was used to map key factors/themes to the domains of the Theoretical Domains Framework.”

Response 5: Thank you, I have revised the methods section to include a detailed description of what was done. This can be found on page 4 line 161-179

Comment 6: What are these descriptive terms in terms of the research methods - attitudes, viewpoints, opinions, and behavior/practice?

“Nine key themes were identified during analysis as key determinants influencing attitudes, viewpoints, opinions, and behavior/practice of health professionals regarding plant-based nutrition.”

Response 6: This paragraph has been revised to “Nine key themes were identified during analysis as determinants influencing health professionals’ perceptions (encompassing attitudes, viewpoints, and opinions) and behavioral practices regarding plant-based nutrition. In this study, perceptions refer to the cognitive and affective factors including attitudes (feelings towards PBDs), viewpoints (broader professional perspectives shaped by experience) and opinions (specific beliefs). Behavioral practices refer to clinical actions and routine behaviors related to recommending and prescribing PBDs in patient care.” This can be found on page 16, line 229-235.

Comment 7: The following doesn’t match with the study’s research question.

“The purpose of this review was to assess health professionals' attitudes and perceptions toward plant-based nutrition, and how these influence their decision to incorporate it into their routine practice.”

Research Question: This review seeks to explore an important research question: what factors serve as enablers and barriers for health care professionals regarding their recommendation of plant-based nutrition to their patients?

Response 7: Thank you for pointing this out. This has been revised to match the research question as follows: “This review assessed health professionals' attitudes and perceptions and the key barriers and enablers influencing their integration of PBDs into routine patient care, with the overarching goal of informing strategies to promote wider adoption of plant-based nutrition among Americans.” This can be found on page 21 line 347-349.

Comment 8: Examples of these should be provided along with references.

“Access to up-to-date, evidence-based guidelines and resources will further empower health professionals to confidently recommend plant-based nutrition, ensuring that patient care is informed by the latest scientific research.”

Response 8: Examples with references have been provided on page 23 line 456-463

Comment 9: This sentence is too dense, which makes it hard to understand. 

“To further encourage the shift towards adopting PBDs, a holistic approach is required that integrates regulatory, technological, financial, and environmental factors [11, 104]. “

Response 9: This sentence was removed from the discussion section.

Comment 10: This needs to be rewritten. What are the limitations of this review study? What could be done in the next study to overcome the limitations?

“The predominance of observational studies in the analysis limits the generalizability of the findings. However, inclusion of studies from various countries with various health professionals provided a more diverse background. Future research could address some of these limitations by incorporating a larger number of studies.”

Response 10: This paragraph has been revised to “The predominance of observational studies in the analysis limits the generalizability of the findings. Although inclusion of studies from multiple countries and a variety of health professionals provides a more diverse perspective, the overall evidence remains limited by study design. To overcome these limitations, future studies should aim to incorporate a greater number of studies, including longitudinal studies to enhance robustness and applicability of results.” 

This can be found on page 24, line 480-485.

Thank you!

Response to Reviewer 1 Comments

Thank you very much for taking the time to review our manuscript. Please find the detailed responses below and the corresponding revisions/corrections highlighted in red in the re-submitted file.

Comment 1: The mapping mentioned in the abstract isn’t described in the methods section.

“These themes were mapped to the domains of the Theoretical Domains Framework (TDF) to identify key enablers and barriers to implementation of PBDs in routine care for patients.”

Response 1: Thank you for pointing this out. I agree with this comment. Therefore, I have revised the methods section to include a detailed description of what was done. This can be found on page 4 line 161-179

Comment 2: The initial search? When were other searches performed?

“The initial search was done on April 18, 2024.”

Response 2: Agree. I have, accordingly modified the sentence to “The search was done on April 18, 2024.” This can be found on page 4 line 143.

Comment 3: References needed for Scispace and Bunni.

Response 3: references have been provided on page 5, line 185 and lines 638 and 639.

Comment 4: What are vegan/plant-based health professionals?

“Elimination criteria included: articles not written in English, commentaries or reviews, studies where only vegan/plant-based health professionals were involved, studies where intervention preceded assessment, and studies where subjects were not health professionals.”

Response 4: This has been clarified to “health professionals following a vegan or plant-based diet,” on page 4, line 157

Comment 5: The following needs to be flushed out in further detail. Methods need to be described in enough detail so that others could replicate the study.

“A cross-study comparison was conducted to validate and refine these preliminary factors/themes. Factors identified within individual studies were cross-referenced with themes observed across multiple studies to identify overarching patterns. This process of comparison and synthesis allowed for the development of comprehensive themes/factors that captured the breadth and depth of the research question. A combined deductive/ inductive approach was used to map key factors/themes to the domains of the Theoretical Domains Framework.”

Response 5: Thank you, I have revised the methods section to include a detailed description of what was done. This can be found on page 4 line 161-179

Comment 6: What are these descriptive terms in terms of the research methods - attitudes, viewpoints, opinions, and behavior/practice?

“Nine key themes were identified during analysis as key determinants influencing attitudes, viewpoints, opinions, and behavior/practice of health professionals regarding plant-based nutrition.”

Response 6: This paragraph has been revised to “Nine key themes were identified during analysis as determinants influencing health professionals’ perceptions (encompassing attitudes, viewpoints, and opinions) and behavioral practices regarding plant-based nutrition. In this study, perceptions refer to the cognitive and affective factors including attitudes (feelings towards PBDs), viewpoints (broader professional perspectives shaped by experience) and opinions (specific beliefs). Behavioral practices refer to clinical actions and routine behaviors related to recommending and prescribing PBDs in patient care.” This can be found on page 16, line 229-235.

Comment 7: The following doesn’t match with the study’s research question.

“The purpose of this review was to assess health professionals' attitudes and perceptions toward plant-based nutrition, and how these influence their decision to incorporate it into their routine practice.”

Research Question: This review seeks to explore an important research question: what factors serve as enablers and barriers for health care professionals regarding their recommendation of plant-based nutrition to their patients?

Response 7: Thank you for pointing this out. This has been revised to match the research question as follows: “This review assessed health professionals' attitudes and perceptions and the key barriers and enablers influencing their integration of PBDs into routine patient care, with the overarching goal of informing strategies to promote wider adoption of plant-based nutrition among Americans.” This can be found on page 21 line 347-349.

Comment 8: Examples of these should be provided along with references.

“Access to up-to-date, evidence-based guidelines and resources will further empower health professionals to confidently recommend plant-based nutrition, ensuring that patient care is informed by the latest scientific research.”

Response 8: Examples with references have been provided on page 23 line 456-463

Comment 9: This sentence is too dense, which makes it hard to understand. 

“To further encourage the shift towards adopting PBDs, a holistic approach is required that integrates regulatory, technological, financial, and environmental factors [11, 104]. “

Response 9: This sentence was removed from the discussion section.

Comment 10: This needs to be rewritten. What are the limitations of this review study? What could be done in the next study to overcome the limitations?

“The predominance of observational studies in the analysis limits the generalizability of the findings. However, inclusion of studies from various countries with various health professionals provided a more diverse background. Future research could address some of these limitations by incorporating a larger number of studies.”

Response 10: This paragraph has been revised to “The predominance of observational studies in the analysis limits the generalizability of the findings. Although inclusion of studies from multiple countries and a variety of health professionals provides a more diverse perspective, the overall evidence remains limited by study design. To overcome these limitations, future studies should aim to incorporate a greater number of studies, including longitudinal studies to enhance robustness and applicability of results.” 

This can be found on page 24, line 480-485.

Thank you!

Reviewer 2 Report

Comments and Suggestions for Authors

Dear authors,

The topic addressed in this review is of great importance and relevance in the context of the aging population, the rise of non-communicable chronic diseases prevalence, and the negative impact of climate change on human and planetary health.
From the analysis of the paper provided, I would like to do a few observations:

  1. 1. The studies included in the analysis assess the level of knowledge, attitudes, and perceptions regarding Plant-Based Diets (PBDs) among various professional categories within the medical field, each with different responsibilities, training, and expertise in nutrition. Thus, it is self-evident that there will always be differences in the barriers to providing nutritional information to patients. The physician is the one who recommends a certain dietary patternn or a restrictive diet, while the dietitian/nutritionist, due to their specialized training, is the appropriate professional to guide and educate the patient in order to achieve maximum therapeutic benefits and ensure long-term compliance.
    For this reason, I believe that a separate analysis of the studies included in the review by professional category—doctor vs. dietitian/nutritionist, etc.—would contribute to obtaining clearer results and formulating conclusions based on the actual situation.
  2. In the discussion section, it would be beneficial to mention that there are few professional societies that have recently updated (2023) their nutritional recommendations based on plant-based diets. Therefore, it is somewhat expected that healthcare personnel may lack knowledge regarding PBDs or sustainable diets. Even though the US Dietary Guidelines for 2020-2025 encourage the consumption of plant-based foods, there are significant differences between the consumption recommendations for the main food groups in these guidelines and the EAT-Lancet Commission or WHO regarding Sustainable Diet .
    There is a need to update the healthy eating guidelines taking into account the benefit of plant-based diets resulting from scientific research, so that physicians have access to current scientific information and can be confident that what they recommend will benefit their patients.
    What I propose at this point is to analyze the barriers and drivers for recommending a plant-based diet separately for physicians versus nutritionists/dietitians. In this way, the findings
    will be able to support specific actions aimed at improving the training of medical staff and hence the quality of health care and nutritional therapy.

Author Response

Response to Reviewer 2 Comments

Thank you very much for taking the time to review our manuscript. Please find the detailed responses below and the corresponding revisions/corrections highlighted in red in the re-submitted file.

Comment 1: The studies included in the analysis assess the level of knowledge, attitudes, and perceptions regarding Plant-Based Diets (PBDs) among various professional categories within the medical field, each with different responsibilities, training, and expertise in nutrition. Thus, it is self-evident that there will always be differences in the barriers to providing nutritional information to patients. The physician is the one who recommends a certain dietary pattern or a restrictive diet, while the dietitian/nutritionist, due to their specialized training, is the appropriate professional to guide and educate the patient in order to achieve maximum therapeutic benefits and ensure long-term compliance.
For this reason, I believe that a separate analysis of the studies included in the review by professional category—doctor vs. dietitian/nutritionist, etc.—would contribute to obtaining clearer results and formulating conclusions based on the actual situation.

Response 1: Thank you for pointing this out. I agree with this comment. Therefore, I have revised the Results section to include a paragraph and a table which includes a subgroup analysis of enablers and barriers by professional category (physicians/clinicians versus dietitians/nutritionists) (table 4) on pages 5 and 6, lines 337-345. I have also included a paragraph in the discussion section on this on page 23, lines 421-431.

Comment 2: In the discussion section, it would be beneficial to mention that there are few professional societies that have recently updated (2023) their nutritional recommendations based on plant-based diets. Therefore, it is somewhat expected that healthcare personnel may lack knowledge regarding PBDs or sustainable diets. Even though the US Dietary Guidelines for 2020-2025 encourage the consumption of plant-based foods, there are significant differences between the consumption recommendations for the main food groups in these guidelines and the EAT-Lancet Commission or WHO regarding Sustainable Diet .
There is a need to update the healthy eating guidelines taking into account the benefit of plant-based diets resulting from scientific research, so that physicians have access to current scientific information and can be confident that what they recommend will benefit their patients.

Response 2: This has been addressed on page 23 lines 425-

                                                              Thank you!
